# The Regularities of Metal Transfer by a Nickel-Based Superalloy Tool during Friction Stir Processing of a Titanium Alloy Produced by Wire-Feed Electron Beam Additive Manufacturing

Valery Rubtsov *, Andrey Chumaevskii *, Evgeny Knyazhev, Veronika Utyaganova, Denis Gurianov, Alihan Amirov, Andrey Cheremnov and Evgeny Kolubaev

Institute of Strength Physics and Materials Science, Siberian Branch of Russian Academy of Sciences, 634055 Tomsk, Russia; clothoid@ispms.ru (E.K.); filaret_2012@mail.ru (V.U.); gurianov@ispms.ru (D.G.); amirov@ispms.ru (A.A.); amc@ispms.ru (A.C.); eak@ispms.ru (E.K.)

* Correspondence: rvy@ispms.ru (V.R.); tch7av@ispms.ru (A.C.)

**Abstract:** In this work, the interaction of an additively produced Ti-$_4$Al-$_3$V titanium alloy with a nickel superalloy tool and the features of the stir zone formation during friction stir processing have been studied. The stop-action technique was used to produce the samples to be studied using optical and scanning electron microscopy methods, as well as microhardness measurements. As a result, it was revealed that the tool, when moving, forms a pre-deformed area in front of it, which is characterized by a fine-grained structure. The presence of an interface layer between the workpiece material and primary fragmentation by the tool was revealed. It was demonstrated that the transfer of titanium alloy material occurs periodically following the ratio of feeding speed to tool rotation rate. Metal flow around the tool can occur in both laminar and vortex modes, as indicated by the tool material stirred into the transfer layer and used as a marker.

**Keywords:** friction stir processing; additive manufacturing; wire; titanium alloy; material transfer; microstructure; microhardness



## 1. Introduction

Friction stir welding and processing (FSW/FSP) of titanium alloys is currently becoming an increasingly important and frequently used technology. This is explained by the fact that titanium alloys remain the most used in many industries, ranging from the production of mobile device bodies to aerospace components [1–3]. In addition, the growing demand for additive technologies is also contributing to the increasing popularity of FSW/P of titanium alloys, as the challenge of improving the structure and mechanical properties of additive products is now acute [4–6]. This problem is most serious in the case of directed energy deposition methods to produce large-sized titanium alloy products [7]. This is primarily caused by the complex thermal cycles in the 3D printing process, which leads to the formation of a directionally oriented coarse-grained structure [8,9]. In the case of titanium alloys, the structure is represented by columnar grains of the primary β phase, and the length of these grains can reach several centimeters [10]. At the same time, the size and morphology of the α and α′ phase plates, which fill the columnar grains, change with the growth of the additive product wall due to changes in the heat dissipation conditions [11]. The above features lead to a high anisotropy of both the structure and mechanical characteristics of the titanium alloy material [12]. Moreover, the properties of additive alloys, in this case, are significantly inferior to the required values [13]. In turn, friction stir processing, which allows for modifying the structure of both the surface and the volume of the product due to dynamic recrystallization, can be used to locally improve the characteristics of additive parts and has already shown positive results both in the processing of aluminum alloys [14,15] and titanium alloys [16,17]. However, in the case of

titanium alloys, the process of the stir zone formation by FSW/P and the peculiarities of material flow are still underexplored.

The stop-action technique [18] is the most suitable for studying the deformation peculiarities and the material structure formation in the FSW-tool effect zone. It allows us to simultaneously study both the structure of the workpiece and the features of tool wear during FSW/FSP and has found application even in the field of steel welding [19]. This method consists of abruptly interrupting the welding/processing process so that the tool remains in the body of the welded workpiece. The resulting sample is then cut into sections, and the structure of the workpiece around the tool is examined. This approach also provides an opportunity to evaluate the fragmentation and adhesion transfer features of the material during FSW/FSP, as shown in [20]. This will help us to better understand the nature of the titanium alloy interaction with the tool and the formation of the stir zone. Thus, the purpose of this work is to study the material structure of additive Ti-4Al-3V alloy in the zone of tool influence using the stop-action technique and to identify the features of the stir zone formation during friction stir processing.

## 2. Materials and Methods

To study the peculiarities of the stir zone formation, samples of wall-shaped workpieces made of Ti-$_4$Al-$_3$V titanium alloy (analog widely used in industry Ti-$_6$Al-$_4$V alloy) were produced by the wire-feed electron beam additive manufacturing method. Three-dimensional printing was performed on a self-developed wire-feed EBAM machine at the Institute of Strength Physics and Materials Science, Tomsk, Russia. The workpieces in as-built condition had dimensions of 300 mm in height, 100 mm in length, and 6 mm in width. Before friction stir processing, 2.5 mm thick plates were cut out of these blanks by EDM-machine DK7750 (Suzhou Simos CNC Technology Co., Ltd., Suzhou, China), similar to the scheme presented in [17]. After that, friction stir processing was performed on a homemade friction stir welding machine using the following parameters: loading force 33–34 kN, tool travel speed 90 mm/min, and tool rotation rate 400 rpm. The workpieces were processed in two directions: vertical (along the wall growth direction) and horizontal (along the layer deposition direction). During the processing, an interruption was performed so that the tool remained in the workpiece according to the stop-action technique. Then, the obtained sample was sectioned using EDM-machine DK7750 according to the scheme shown in Figure 1.

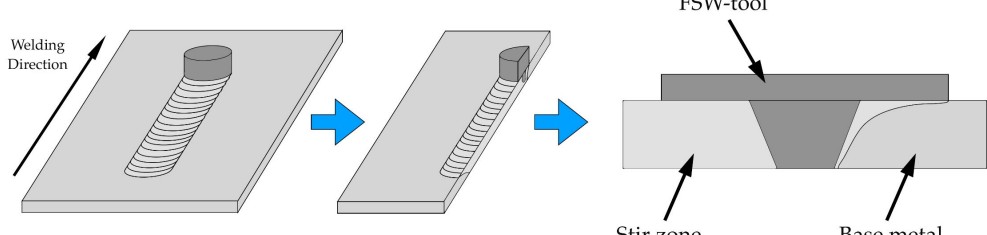

**Figure 1.** Schematic illustration of sample cutting.

The resulting sections were mechanically ground, polished, and chemically etched to study the structure of the material. The structure was analyzed using the Altami MET-1C optical microscope (Altami Ltd., Saint Petersburg, Russia). In addition, studies were carried out using scanning electron microscopy for finer microstructure and elemental analyses. For this purpose, the Apreo 2 S LoVac scanning electron microscope (Thermo Fisher Scientific Inc., Waltham, MA, USA) was used. The microhardness of the material was measured in a longitudinal section with a load of 0.5 kg and an increment of 0.5 mm. Duramin 5 microhardness tester (Struers A/S, Ballerup, Denmark) was used for this procedure. Mechanical tests were performed on a universal testing machine UTS 110M (Testsistems, Ivanovo, Russia).

## 3. Results

As a result of the experiments, two samples with the tool remaining in the workpiece after processing in the horizontal and vertical directions were prepared. First, the macrostructure of the samples was investigated (Figure 2). The specimens showed a distinct pre-deformed area (PDA) in the region in front of the tool, which borders the base metal. Light-colored interfaces are identified on the tool predominantly in the lower half of the pin, but they are also observed in the area under the tool shoulders on the stir zone side (SZ, Figure 2b). The shape of this surface layer in the horizontal sample, as well as the shape of the worn tool in combination with the visible part torn off and mixed into the stir zone surface layer in the vertical sample, may indicate that the formation of this layer is one of the mechanisms of tool wear during FSW/P of titanium alloys. This can also be evidenced by the periodic mixing of this material in the SZ, which is observed in the lower part of the workpiece. It should be noted that in the vertical sample, the tool was worn more intensively, and therefore, the stirring of the interface layer in the SZ was also more intense compared to the horizontal sample. This condition may be related to the mechanical material properties of the additively produced samples, which will be described later.

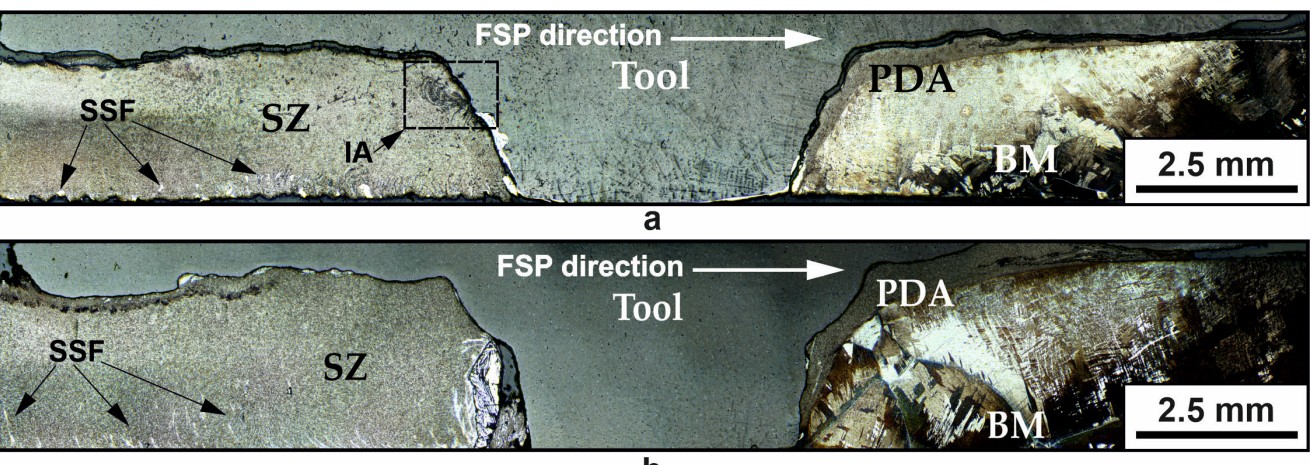

**Figure 2.** Macroscopic images of samples produced by stopping FSP process when processing along horizontal (**a**) and vertical (**b**) directions. SZ—stir zone; PDA—pre-deformed area; BM—base metal; SSF—stainless steel fragments; IA—area of interaction between tool and material.

Figure 3 shows images of the microstructure of the sample produced by processing in the horizontal direction. The microstructure of the base metal (Figure 3a) presents a typical structure for additive material and represents grains of primary β phase separated by grain-boundary α phase (α-GB). Inside these grains are large colonies of alpha-phase plates. After processing, due to severe plastic deformation and dynamic recrystallization, the structure of the material changes from lamellar to globular with small, almost equiaxed grains in the stir zone (Figure 3c). Of greatest interest is the pre-deformed area in the region in front of the tool (Figure 3b). Its dimensions are of the order of 200 μm and vary with the height of the workpiece. The greatest width of the pre-deformed area is observed in the transition zone from the shoulder part of the tool to the pin, i.e., where the strongest tangential stresses during machining occur [21]. The interface layer observed in the macro-images at different parts of the sample has different sizes and predominantly has a layered structure due to the high deformation degree at FSP (Figure 3c). The largest proportion of this layer is concentrated in the lower part of the pin on the side behind the tool.

In the sample produced by processing in the vertical direction, the structure is formed similarly. The base metal is also represented by primary β-phase grains and α colonies (Figure 3d), and the stir zone is represented by small recrystallized α grains (Figure 3f). However, in contrast to the horizontal sample, the interface layer between the tool and the workpiece has a longer extent, and, in addition to the layered structure, it also has a

fragmented structure (Figure 3f). The average grain size is about 4.9 ± 1.5 μm. The size of the transition zone is similar to the previous example, but since the tool is more worn here, in the pre-deformed area, there are inclusions of surface layer particles displaced as a result of tool wear during processing (Figure 3e). The grain size in the pre-deformed area is smaller than in the stir zone and is 3.3 ± 1.4 μm, which is also confirmed by the darker etching in the macro-image due to the higher density of grain boundaries.

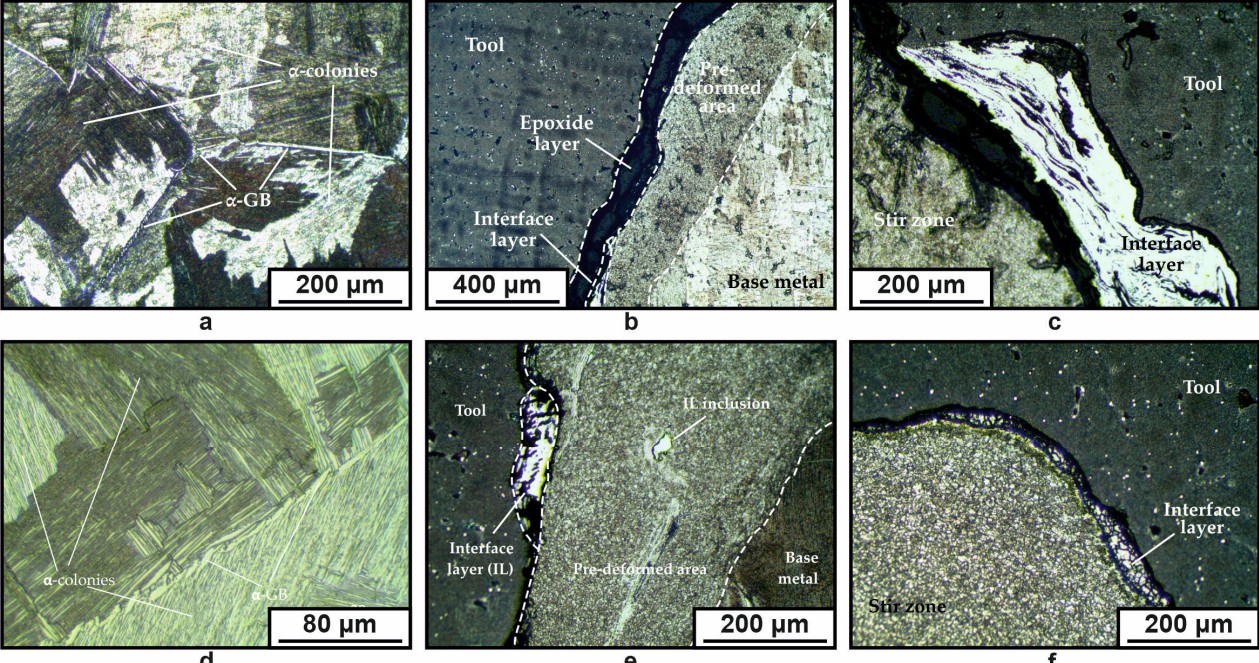

**Figure 3.** Metallography images of the base metal (**a**,**d**), area in front of the tool (**b**,**e**), and area behind the tool (**c**,**f**) in the sample produced along horizontal (**a**–**c**) and vertical (**d**–**f**) directions. GB—grain boundary.

The microhardness measurement shows that the maximum value of hardness in each of the specimens falls on the interface between the tool and the material of the stir zone, which exceeds the hardness of the tool (Figure 4a,b). It should be noted that the microhardness in the stir zone is slightly higher than in the base metal. The increase in hardness of the stir zone is caused by grain refinement, and the boundary between it and the tool is formed by intermetallic phases in the tribological layer during processing. At the same time, after processing, regardless of the initial microhardness of the material and the direction of processing, the microhardness is at the same level and becomes much more stable compared to the base metal. This shows, on the one hand, that material hardening takes place and, on the other hand, that the mechanical properties of the formed structure do not depend significantly on the initial state before processing.

The mechanical properties of the sample material before processing differ significantly in the vertical and horizontal directions (Figure 4c). This is particularly pronounced for the strength and ductility of materials in compression tests. After processing in both tension and compression, the material properties of the stir zone increase significantly and are much less dependent on the initial orientation of the specimens. At the same time, due to the penetration of tool and substrate fragments into the material of the stir zone in tension, the specimens are characterized by lower plasticity values after processing, while in compression, the plasticity of the specimens remains at a relatively high level.

Increased tool wear during vertical processing can be attributed to the higher compressive strength and ductility values of the material for these specimens. This results in higher stresses in the tool–material contact zone during processing compared to specimens processed in the horizontal direction.

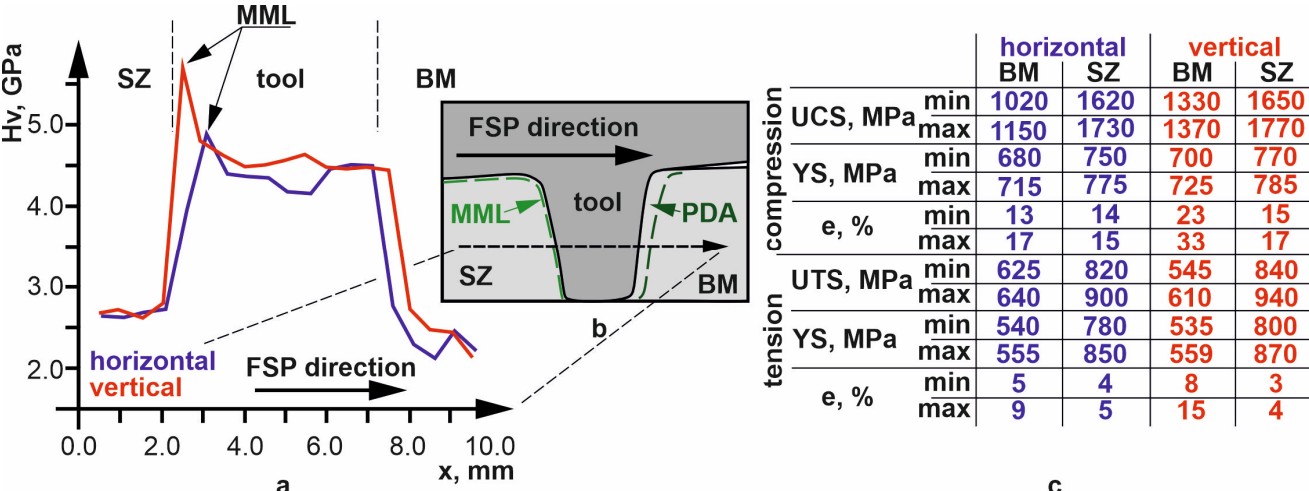

|  |  |  | horizontal BM | horizontal SZ | vertical BM | vertical SZ |
|---|---|---|---|---|---|---|
| compression | UCS, MPa | min | 1020 | 1620 | 1330 | 1650 |
|  |  | max | 1150 | 1730 | 1370 | 1770 |
|  | YS, MPa | min | 680 | 750 | 700 | 770 |
|  |  | max | 715 | 775 | 725 | 785 |
|  | e, % | min | 13 | 14 | 23 | 15 |
|  |  | max | 17 | 15 | 33 | 17 |
| tension | UTS, MPa | min | 625 | 820 | 545 | 840 |
|  |  | max | 640 | 900 | 610 | 940 |
|  | YS, MPa | min | 540 | 780 | 535 | 800 |
|  |  | max | 555 | 850 | 559 | 870 |
|  | e, % | min | 5 | 4 | 8 | 3 |
|  |  | max | 9 | 5 | 15 | 4 |

**Figure 4.** Vickers microhardness (HV) of samples (**a**), schematic of its measurement area (**b**), and mechanical properties of the material after printing and after processing (**c**).

Scanning electron microscopy was used to study the surface layer in different parts of the tool, including measurements of the chemical composition of this layer. It was revealed that the coarse layers in the lower part of the tool and the continuous layers in the middle of the pin and under the shoulders are different. The first type of interface layer (Figure 5) is a continuous layer formed as a result of the interaction between the tool and the workpiece. As the analysis of chemical composition shows, this layer is based on titanium alloy (76% Ti, 6.5% Al, 3.3% V) with the inclusion of elements of the tool material (9.4% Ni, 2.3% Cr, 1.6% Co, and 0.9% W). Thus, it can be concluded that this layer is formed as a result of the diffusion of nickel alloy elements into the workpiece material, probably in the process of adhesion–diffusion wear of the tool. As for the structure of this layer, the image in Figure 5 shows that this layer is fragmented into small cells near the tool and larger cells in the area of contact with the workpiece material (1). There is a clear boundary between the surface layer and the stir zone (2). In the area near the boundary, there are also traces of further fragmentation deep into the workpiece (3).

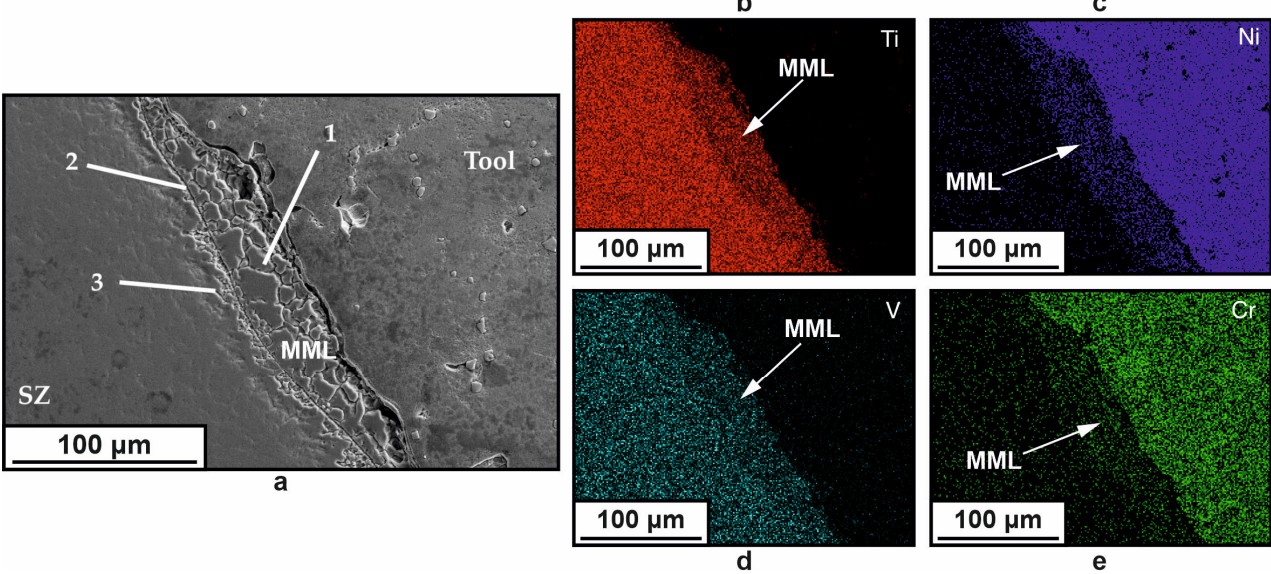

**Figure 5.** SEM image of the first type of interface layer between tool and stir zone (**a**) and maps of the chemical element distribution (**b–e**).

The second type of interface layer, which is observed predominantly at the bottom of the tool and mixed at the bottom of the stir zone, is shown in Figure 6. Chemical analysis shows that the material of this layer is not related to either the workpiece material or the tool. The chemical element distribution maps (Figure 6d,e) showed that this layer is chromium–nickel steel. Thus, the tool may have encountered the substrate during the welding/processing operation. Although this is not the correct course of the FSP process, it allows us to demonstrate the adhesive nature of the stir zone formation using steel as a marker material. First, the steel material is captured by the bottom plane of the pin and is carried upward, reaching the middle of the pin along the side (Figure 2). Then, as the tool rotates and moves, this layer is detached from the pin and mixed into the workpiece material. The material transfer process is periodic but irregular, as can be seen in Figure 2.

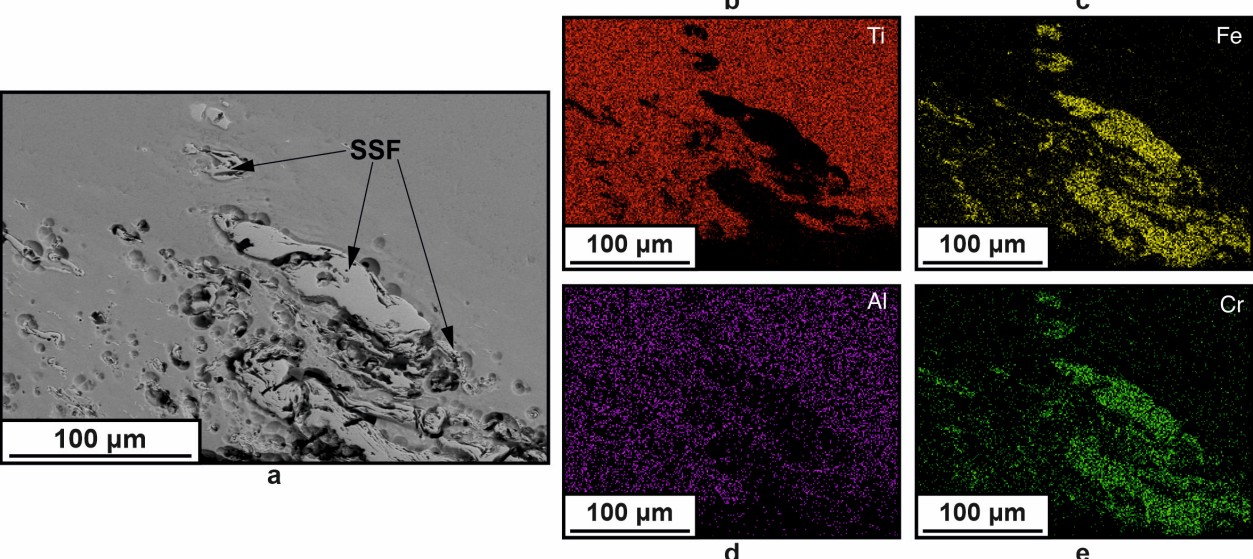

**Figure 6.** SEM image of the second type of interface layer mixed into the stir zone (**a**) and maps of the chemical element distribution (**b**–**e**).

## 4. Discussion

Studies have shown that the tool, during friction stir processing, significantly deforms the material in front of it while moving. As the analysis of the material microstructure showed, the size of the pre-deformed area is about 200 microns, and the grain size in it is, on average, 3.3 μm. When the material is transferred to the zone behind the tool, i.e., forms the stir zone, the material structure recrystallizes, as a result of which the grain size in the stir zone increases and averages 4.9 μm. At the same time, in the contact zone of the tool with the workpiece, there is a layer of material representing a titanium alloy with a small share of chemical elements of the tool material. Based on the structure of this layer (Figures 4 and 5), it could be formed because of the primary fragmentation of the material, i.e., at the initial stage of the transfer layer formation. From the data obtained, it can be concluded that, as a result of thermomechanical action, the tool fragments a material layer up to 30 μm thick, depending on the tool section and the stresses in this section. Part of this layer eventually forms a transfer layer, while another part recrystallizes with the pre-deformed area material.

Once the transfer layer reaches a critical size, it is adhered to the tool and transferred to the area behind the tool. By stirring a foreign material (stainless steel), it was possible to trace the material transfer pattern during friction stir processing of titanium alloy. It is known that an approximate estimate of the thickness of the transferred layers can be made through the ratio of the tool feeding speed to the tool rotation rate. In this case, at a travel speed of 90 mm/min and a rotation rate of 400 rpm, the tool passes 0.225 mm per rotation. Consequently, the tool can transfer a material layer with a thickness of about 0.225 mm per

revolution. Using optical metallography images, the distances between the mixed layers of steel were measured (Figure 7).

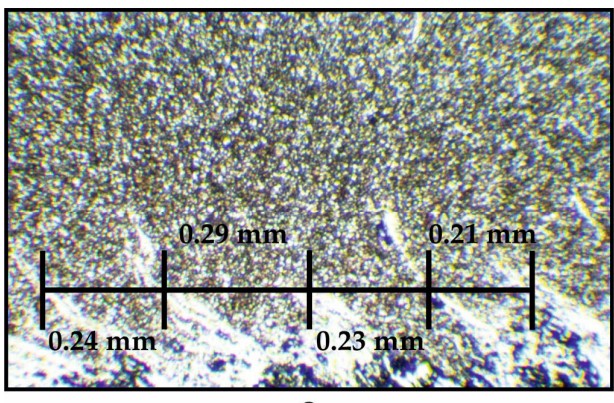 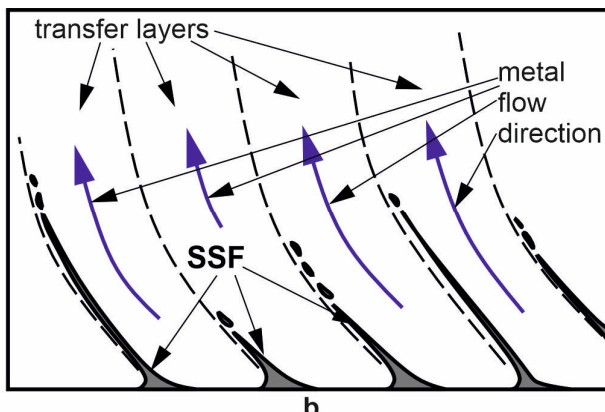

**Figure 7.** Microscopic image illustrating the pattern of material stirring with distances between intermixed layers (**a**) and a schematic representation of the steel fragments' penetration process into the stir zone (**b**).

The measurements showed that the distance between the layers of the steel mixed into the stir zone is about $0.24 \pm 0.034$ mm. Therefore, since stainless steel acts as a marker material, it can be stated that the thickness of the transferred layers is almost equal to the calculated value of tool travel per rotation. Consequently, for titanium alloys, the regularities that have been determined for welding aluminum alloys are valid both in terms of material fragmentation and the formation of transfer layers and in terms of the regularities of the stir zone formation, depending on the process parameters.

Stirring of stainless steel fragments from the substrate takes place on each of the transfer layers, alternating in the stirring zone in the longitudinal section (Figure 7b). In the transfer layers, due to the external forces and the characteristics of the flow organization along the contour of the tool, the material is stirred both in the horizontal plane and in the vertical direction. In the vertical direction, the flow of metal is initiated as a reaction of the material to the force exerted by the tool due to the force of normal pressure on the workpiece. As a result, in case of sufficiently close contact in the lower part of the tool and the substrate, its surface is deformed, plasticized, and pressed into the material of the stir zone, simultaneously undergoing deformation and structural phase interaction with the flows of the base material. In addition, due to the complex nature of the metal flow in the metal streams, the substrate fragments are crushed and distributed more evenly in the stir zone.

It should be noted that the friction stir process has an extrusion component in addition to the adhesive component. This is due to the pressure of the tool on the material in front of it, which is in a deformed to fine-grained state and at a temperature of 0.6–0.8 of the melting point. The material is then carried behind the tool by adhesion and by superplastic deformation or extrusion. As shown in the research data, the material movement also has a vertical component, even under significant tool loads when processing titanium alloys. This results in the stirring of the substrate material to a considerable height from the bottom of the workpiece. At the same time, there is a structural phase interaction between the surface layers of the tool and the transferred material, which leads to gradual tool wear and changes in the composition of the stir zone. The interaction of nickel from the tool and titanium leads to the formation of complexly organized intermetallic layers on the tool surface. At the same time, an intermediate layer with a carbide network [22] is formed, which has a higher wear resistance than a layer with a pure intermetallic structure. The process of tool wear, in this case, is practically identical to the previously studied process of adhesive friction and wear with nanostructuring of the surface, the formation of „welding bridges", the formation of a mechanically mixed MML layer from fragments

of parts forming tribo-joint, etc. [23,24]. In such a case, the MML layer on the tool surface is constantly growing due to the continuous diffusion of titanium. At the same time, intermetallic particles mixed into the stirring zone material are constantly chipping off the surface of such a layer due to contact with metal flows. Catastrophic wear is also possible when the overgrown MML detaches from the surface, leaving the tool material exposed to the process of friction and wear for a short time [22]. Studies show that this leads to increased intensity of tool wear in the pin and shoulder contact area [22].

The normal process of mutual reaction between tool and material occurs continuously and directly during welding or friction stir processing [22]. The present work also emphasizes the formation of tool fragments mixed in the stirring zone. The application of the stop-action technique makes it possible to follow the characteristic features of this process in sufficient detail (Figure 8).

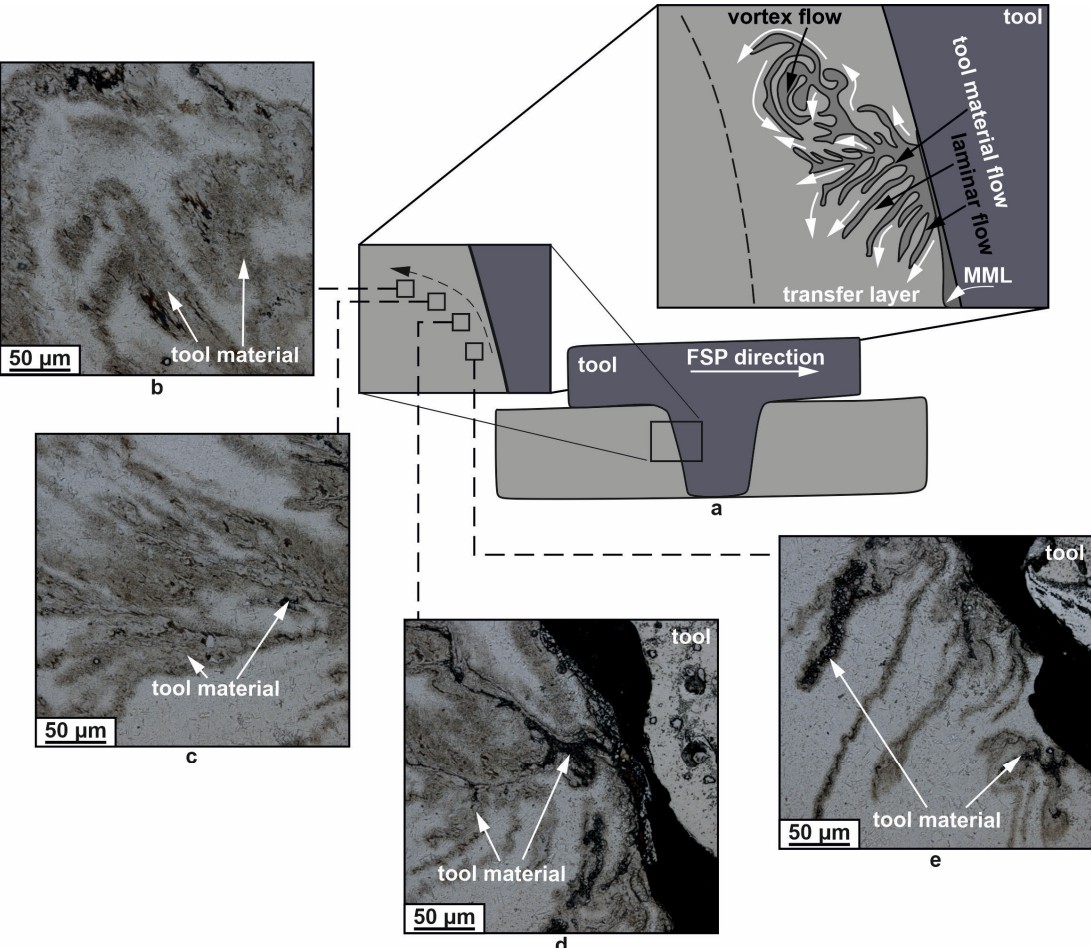

**Figure 8.** Schematic diagram of the tool–material interaction process in transfer layer flow (**a**) and microscopic images of characteristic flow areas (**b**–**e**). The area is marked IA in Figure 2a.

Behind the tool in the zone of the transfer layer, tool fragments are clearly visible (Figure 8a). More specifically, these are fragments of tool material that have mechanically and diffusively stirred with the titanium alloy. For the sake of simplicity, they will be referred to simply as tool fragments. The flow of metal along the contour of the tool is manifested, as mentioned above, in the form of flows of adhesive and is extrusive in nature. Adhesive or adhesive–cohesive flows are predominantly found in the layers closest to the tool. The wider layers, which form the transfer layer, are characterized more by extrusion flow due to the superplastic deformation of the ultrafine-grained material.

In the transfer layer, the metal flow is divided into a horizontal flow around the tool and a vertical flow. The first component is caused by the pressure of the tool as it moves

along the processing line on the deformed and heated material. The second component is caused by the pressure exerted by the tool on the material along the vertical axis. In addition, the tool pressure on the workpiece surface is significantly higher than the reaction of the material to its longitudinal movement. As a result, the metal flow along the tool contour follows a complex trajectory. In the area behind the tool, it is possible to distinguish a zone where the stirring of tool fragments is particularly pronounced (Figure 8a–e).

In this case, near the tool surface (Figure 8d,e), large fragments are separated and stirred deep into the metal, forming elongated lines. In addition, the trajectory of metal movement in such formations can be both horizontal and upward or downward relative to the position of the flow initiation point (Figure 8). The greater the distance from the point of contact between the tool and the material, the less uniform these formations are, and the more curved trajectories with vortex structures are formed (Figure 8b,c). This situation can be caused by the inhibition of the most distant material layers behind the tool due to their interaction with the previously transferred material. The inhibition of the material of the transfer layer leads to the realization of vortex modes of metal flow and stratification of the transfer layer into different regions of metal flow mechanics. In general, the metal flow of a complexly organized type is characteristic of the processing of both sheet metal and additively manufactured products studied previously [25]. When considering the features of the interaction between tool and material in work [22], it can also be observed that in different regions of the transfer layer, the material of the stir zone can exhibit both laminar and vortex structure. Despite the fact that friction stir processing of titanium alloy shows tendencies of intensive diffusion interaction with tools made of cobalt [26], nickel [22,27], boron nitride [28], hard alloys [29], and many different materials [30], the mechanism of metal transfer along the tool contour remains unchanged, and the use of stop-action technique allowed us to visualize its features in more detail in this work.

## 5. Conclusions

The following conclusions have thus been drawn from the research conducted:

1. During friction stir processing of additive Ti-$_4$Al-$_3$V titanium alloy, a wide zone of pre-deformed material is formed in the zone in front of the tool, which is characterized by finer grains than in the stir zone. When the material is transferred behind the tool, the grain size increases due to recrystallization;
2. An interface layer is formed between the tool and the workpiece, which is a layer of primary fragmented material characterized by the diffusion of chemical elements of the tool material into the body of the workpiece;
3. When the tool contacts the substrate, the substrate material is transferred from the bottom to the top and is mixed into the stir zone of the titanium alloy. This material forms a layer between the tool and the workpiece and increases tool wear;
4. The frequency of material transfer during friction stir processing of titanium alloy is determined by the ratio of the tool feeding speed to its rotation rate. In this case, the metal flow on each of the transfer layers is uneven, as evidenced by the different penetration of substrate fragments into the stir zone;
5. Metal flow along the tool contour, visualized by the penetration of tool wear particles into the stir zone, shows tendencies to both laminar and vortex modes;
6. Higher values of mechanical properties in vertical compression of the sample material resulted in more significant tool wear during processing;
7. The mechanical properties of the material of the stir zone after processing are quite similar, with initially significant differences in the horizontal and vertical directions, indicating a decrease in the anisotropy of the properties of the initially additively obtained samples.

**Author Contributions:** Conceptualization, V.R. and A.C. (Andrey Cheremnov); methodology, A.C. (Andrey Chumaevskii); investigation, E.K. (Evgeny Knyazhev), V.U., D.G., A.A. and A.C. (Andrey Cheremnov); resources, A.C. (Andrey Chumaevskii); data curation, A.C. (Andrey Cheremnov);

writing—original draft preparation, A.C. (Andrey Chumaevskii); writing—review and editing, V.R.; visualization, E.K. (Evgeny Kolubaev), V.U., D.G. and A.C. (Andrey Cheremnov); project administration, V.R.; funding acquisition, V.R. All authors have read and agreed to the published version of the manuscript.

**Funding:** The work was performed according to the Government research assignment for ISPMS SB RAS, project FWRW-2022-0004.

**Data Availability Statement:** The data presented in this study are available in The Regularities of Metal Transfer by a Nickel-Based Superalloy Tool during Friction Stir Processing of a Titanium Alloy Produced by Wire-Feed Electron Beam Additive Manufacturing.

**Conflicts of Interest:** The authors declare no conflicts of interest.

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
