# Peer review of "The Regularities of Metal Transfer by a Nickel-Based Superalloy Tool during Friction Stir Processing of a Titanium Alloy Produced by Wire-Feed Electron Beam Additive Manufacturing"

_metals, doi:10.3390/met14010105_

Round 1

Reviewer 1 Report

Comments and Suggestions for Authors

In this study, the interaction of additively produced Ti6Al4V titanium alloy with a nickel superalloy tool and the features of the stir zone formation during friction stir processing have been studied. The results are interesting and can be recommended to be published, however, after the following considerations.

1, Please indicate the dimension of the tool and the welding direction in Fig.2. It was reported that the WC-based tool, pcBN and Co alloy tools could be successfully used for FSW of Ti alloys without serious tool wear. It was not clear why the authors chose Ni alloy tools in this study.

2, Why the tool was worn more intensively in the vertical sample than in the horizontal sample? The difference of these two FSW samples should be summarized in the conclusions section.

3, The grain size mentioned in Line 130 and Line 134 was for α phase or β phase?How to measure the grain size in the pre-deformed area and the stir zone? It is impossible to distinguish the grain size in Fig. 3 and Fig. 4.

4, The microstructure of the pre-deformed area and stir zone needed elaborately characterized.

5, Why the hardness of the stir zone was lower than that of the base metal, regardless of the refined microstructure in stir zone?

Author Response

Dear Mr./Ms. reviewer

Thank you very much for your comments

The following are responses to your comments

Best regards

Authors team

In this study, the interaction of additively produced Ti6Al4V titanium alloy with a nickel superalloy tool and the features of the stir zone formation during friction stir processing have been studied. The results are interesting and can be recommended to be published, however, after the following considerations.

1, Please indicate the dimension of the tool and the welding direction in Fig.2. It was reported that the WC-based tool, pcBN and Co alloy tools could be successfully used for FSW of Ti alloys without serious tool wear. It was not clear why the authors chose Ni alloy tools in this study.

Partially agree with the comment. Figure 2 has been revised. WC-based alloys, pcBN and Co tools, according to the analysis of the current state of the problem, are also either subject to wear during welding or processing of titanium, or cause the formation of brittle phases in the stir zone that reduce its mechanical properties (10.3390/met13050970). In general, the wear mechanisms of nickel-based and cobalt-based tools are even quite similar (https://doi.org/10.1016/j.wear.2021.204180, https://doi.org/10.1016/j.wear.2021.204138). In comparison with expensive tools based on W-Re or PcBN alloys, a tool made of nickel heat-resistant alloy is much less expensive, which makes it practical for FSW/FSP titanium alloys. (Твой вариант перевод)

Figure 2 has been revised. Severe wear of welding tools during FSW/FSP of titanium alloys is reason why these processes don’t use in industries (https://doi.org/10.1016/j.jmst.2017.10.018). In case of pcBN tool there are few researches (https://doi.org/10.1016/j.msea.2007.10.062, https://doi.org/10.1016/j.matchemphys.2014.04.002) that demonstrates tool debris in the weld which makes such chemical compounds as TiB, TiB2, Ti2N etc. That compounds makes welds brittle. Therefore, these tools cannot be used during FSW/FSP of titanium alloys, unlike FSW of steels, where pcBN tools showed excellent results. In case of WC-based and Co-based tools which has wear very fast during FSW/FSP of titanium alloys (https://doi.org/10.1179/1362171811Y.0000000023). Unfortunately, none of the current research demonstrates the how long a connection can be made before the tool fails. Based on personal experience, the authors suggest that for a WC-based tool the joint length will be about 700 mm. However, for Ni-based tools, such investigations are available and they demonstrate that the tool produces 1.6 meters of weld without liquid cooling and 2.8-2.9 meters with liquid cooling (https://doi.org/10.3390/met10060799, https://doi.org/10.3390/lubricants11070307). In addition, heat-resistant Ni-based tools are less expensive than all of the tools mentioned above. (Мой вариант).

2, Why the tool was worn more intensively in the vertical sample than in the horizontal sample? The difference of these two FSW samples should be summarized in the conclusions section.

Slightly higher wear intensity of the tool processed titanium alloy in the vertical direction can be related to the higher mechanical properties of the material during compression. This causes higher stresses in the contact zone during processing. For example, the material presented in the article is characterized by the following deformation behavior in compression (graph in the attached file). In this case, samples E are deformed in the vertical direction, and samples D - in the horizontal direction.

This point is further discussed in the study and the results of mechanical tests are added. If necessary, we could add a section with mechanical tests to the paper.

3, The grain size mentioned in Line 130 and Line 134 was for α phase or β phase? How to measure the grain size in the pre-deformed area and the stir zone? It is impossible to distinguish the grain size in Fig. 3 and Fig. 4.

The grain size is referred to the alpha phase in this case. In general, according to X-ray diffraction analysis (DOI 10.3390/ma16113901), titanium alloy Ti-4Al-3V after additive electron beam printing is characterized by a rather low (about 1%) content of beta phase after printing.

4, The microstructure of the pre-deformed area and stir zone needed elaborately characterized.

Agree partially with the comments. But, this work was more aimed at establishing the features of the interaction between the tool and the material. We have discussed the structure formation after processing of the above alloy in more detail in previous publications (doi:10.3390/lubricants10120349, doi:10.3390/met12010055). If you still believe that it is necessary to present the results of more subtle structural studies in this paper, we could add an additional section to the manuscript.

5, Why the hardness of the stir zone was lower than that of the base metal, regardless of the refined microstructure in stir zone?

Thank you very much for the comment. Unfortunately, there was an error in the graphs presented in the paper. The hardness of the mixing zone was higher than in the base metal zone. The hardness of the material of the boundary zone between the tool and the mixing zone is increased due to the formation of intermetallic phases. We have made corrections in the graphs.

Reviewer 2 Report

Comments and Suggestions for Authors

This paper reported the interaction of additively produced Ti6Al4V titanium alloy with a nickel superalloy tool and the features of the stir zone formation during friction stir processing. However, some concerns need to be addressed before the paper can be considered for publication.

1.     The microstructure and phase region calibration may not be accurate enough to rely on an optical microscope alone.

2.     There is a lack of overall consideration of the similarities and differences between the horizontal and vertical directions.

3.     What is the significance of hardness testing?

4.     The surface layer and interface layer calibration should not rely solely on energy spectrum analysis.

Author Response

Dear Mr./Ms. reviewers

Thank you very much for your comments.

The following are responses to your comments.

Best regards

Authors team

This paper reported the interaction of additively produced Ti6Al4V titanium alloy with a nickel superalloy tool and the features of the stir zone formation during friction stir processing. However, some concerns need to be addressed before the paper can be considered for publication.

  1. The microstructure and phase region calibration may not be accurate enough to rely on an optical microscope alone.

In this work, we planned to limit ourselves to describing the interaction between the tool and the material, relying on the results of previous work, where we examined the formation of the structure and phase composition of the resulting materials after 3D printing and FSP (doi:10.3390/lubricants10120349, doi:10.3390/met12010055). If necessary, we could add a section on microstructure studies to this work.

  1. There is a lack of overall consideration of the similarities and differences between the horizontal and vertical directions.

Descriptions of the differences in the structure and properties of the samples after processing in various directions have been added to the manuscript.

  1. What is the significance of hardness testing?

Microhardness measurement shows the differences between the initial state of the material before and after processing in different directions. Tests show that after processing in different directions, the microhardness of the material in the mixing zone, i.e. its mechanical properties are at the same level. Also, microhardness measurements show that its highest values occur in the area behind the tool, in the zone of formation of a mechanically mixed layer of tool material and titanium alloys, where intermetallic phases are formed.

  1. The surface layer and interface layer calibration should not rely solely on energy spectrum analysis.

This work was primarily focused on identifying the characteristics of tool wear during processing and mixing of its material in the stir zone. Additionally, it was revealed that the substrate material can be mixed into the stir zone to a fairly significant depth. In order to show these aspects, information on energy dispersive analysis was carried out. We previously carried out more detailed studies of the microstructure of the mixing zone in other works. If necessary, we could add a section on microstructure studies to this work.

Reviewer 3 Report

Comments and Suggestions for Authors

In my opinion, the presented research is very innovative, interesting, and contains many information crucial to the current state of the art. I have listed a couple of minor remarks to clarify some issues:

1. In the materials and methods part, the Authors refer to the investigated material as “Ti-4Al-3V titanium alloy (analog widely used in industry Ti-6Al-4V alloy” and in the abstract the base material is pointed as “Ti6Al4V”. It should be clarified.

2. In Fig. 6., these steel fragments are the fragments from the upper layer of the backing plate? I am not sure if I understand that part correctly, but we had a similar issue during the FSW of a 3mm Ti6Al4V sheet.

3. Line 171-172 “The material transfer process is periodic but irregular, as can be seen in Figure 2.” What do you mean and where in Figure 2 it can be seen?

Author Response

Dear Mr./Ms. reviewers

Thank you very much for your comments.

The following are responses to your comments.

Best regards

Authors team

In my opinion, the presented research is very innovative, interesting, and contains many information crucial to the current state of the art. I have listed a couple of minor remarks to clarify some issues:

  1. In the materials and methods part, the Authors refer to the investigated material as “Ti-4Al-3V titanium alloy (analog widely used in industry Ti-6Al-4V alloy” and in the abstract the base material is pointed as “Ti6Al4V”. It should be clarified.

Corrected

  1. In Fig. 6., these steel fragments are the fragments from the upper layer of the backing plate? I am not sure if I understand that part correctly, but we had a similar issue during the FSW of a 3mm Ti6Al4V sheet.

This situation is also possible for FSP/FSW of aluminum alloys. In the case of FSP/FSW of titanium alloys, it is more complicated due to the higher temperature in the processing zone and the pressure on the tool, as well as the more intensive interaction between the tool and the material. In a series of experiments, we have also found a similar situation in the processing of sheets of titanium alloys with a thickness of 2 - 3 mm.

  1. Line 171-172 “The material transfer process is periodic but irregular, as can be seen in Figure 2.” What do you mean and where in Figure 2 it can be seen?

In this case, we mean that the substrate material is partially stirred into the stir zone in separate fragments on each of the transfer layers. However, it is embedded at different depths on different layers. The term stirred SSF fragments (stainless steel fragments) has been added to Figure 2.

Round 2

Reviewer 1 Report

Comments and Suggestions for Authors

The revised version is satisfactory and therefore can be accepted for publication.

Reviewer 2 Report

Comments and Suggestions for Authors

after carefully read the revised manuscript, the authors have adressed all my comments, I agree this manuscript for publication